# TAMs and PD-1 Networking in Gastric Cancer: A Review of the Literature

**DOI:** 10.3390/cancers16010196

**Published:** 2023-12-30

**Authors:** Melina Yerolatsite, Nanteznta Torounidou, Aristeidis Gogadis, Fani Kapoulitsa, Panagiotis Ntellas, Evangeli Lampri, Maria Tolia, Anna Batistatou, Konstantinos Katsanos, Davide Mauri

**Affiliations:** 1Department of Medical Oncology, University of Ioannina, 45500 Ioannina, Greece; nadia.torou@gmail.com (N.T.); agogadis@gmail.com (A.G.); fanikap1@yahoo.gr (F.K.); ntellasp@gmail.com (P.N.); dvd.mauri@gmail.com (D.M.); 2Society for Study of Clonal Heterogeneity of Neoplasia (EMEKEN), 45445 Ioannina, Greece; 3Department of Pathology, University of Ioannina, 45500 Ioannina, Greece; elampri@uoi.gr (E.L.); abatista@uoi.gr (A.B.); 4Department of Radiotherapy, University of Crete, 71003 Heraklion, Greece; mariatolia1@gmail.com; 5Department of Gastroenterology, University of Ioannina, 45500 Ioannina, Greece; khkostas@hotmail.com

**Keywords:** tumor microenvironment (TME), tumor-associated macrophages (TAMs), immune checkpoint inhibitors (ICIs), PD-1/PD-L1, gastric cancer

## Abstract

**Simple Summary:**

This study aims to explore the connection between tumor-associated macrophages (TAMs) and the programmed cell death protein 1(PD-1)/programmed death ligand 1 (PD-L1) pathway in gastric cancer (GC). Immune checkpoint inhibitors (ICIs) have shown promise in cancer treatment. Here, we investigate their potential in GC. TAMs, abundant in the tumor microenvironment (TME), can express PD-1, which interacts with PD-L1 on cancer cells. This systematic review indicates that high PD-L1 expression correlates with increased TAM infiltration and may lead to worse patient outcomes. This suggests that TAMs could be crucial in regulating the PD-1/PD-L1 network in GC. Enhanced comprehension of this intricate relationship may offer opportunities for optimizing the efficacy of ICIs in this malignancy.

**Abstract:**

Background: Gastric cancer (GC) is one of the most common and aggressive types of cancer. Immune checkpoint inhibitors (ICIs) have proven effective in treating various types of cancer. The use of ICIs in GC patients is currently an area of ongoing research. The tumor microenvironment (TME) also seems to play a crucial role in cancer progression. Tumor-associated macrophages (TAMs) are the most abundant population in the TME. TAMs are capable of displaying programmed cell death protein 1 (PD-1) on their surface and can form a ligand with programmed death ligand 1 (PD-L1), which is found on the surface of cancer cells. Therefore, it is expected that TAMs may significantly influence the immune response related to immune checkpoint inhibitors (ICIs). Aim of the study: Understanding the role of TAMs and PD-1/PD-L1 networking in GC. Methods: A systematic review of published data was performed using MEDLINE (PubMed), Embase, and Cochrane databases. We retrieved articles investigating the co-existence of TAMs and PD-1 in GC and the prognosis of patients expressing high levels of PD-1+ TAMs. Results: Ten articles with a total of 2277 patients were included in the systematic review. The examined data suggest that the expression of PD-L1 has a positive correlation with the infiltration of TAMs and that patients who express high levels of PD-1+ TAMs may have a worse prognosis than those who express low levels of PD-1+ TAMs. Conclusions: TAMs play a pivotal role in the regulation of PD-1/PD-L1 networking and the progression of GC cells. Nevertheless, additional studies are needed to better define the role of TAMs and PD-1/PD-L1 networking in GC.

## 1. Introduction

Gastric cancer (GC) is one of the most common and aggressive types of cancer. It has the fifth-highest incidence of all cancers and is presently the fourth leading cause of cancer death worldwide [1]. Due to their extensive morphological variability, GC can be classified through various approaches. The widely used tumor–node–metastasis (TNM) classification stratifies it into superficial (T1) or advanced types (T2–T4). Morphological distinctions can also be made based on characteristics such as the presence of mucus in the cytoplasm and the rate of glandular vs. tubular differentiation. Although all gastric adenocarcinomas originate from the stomach’s glandular epithelium, their morphology varies based on the location of affected glands (pyloric, fundic, cardiac), the extent of surface involvement, and potential metastatic lesions [1,2,3,4].

Moreover, histologically, GC has different subtypes, and for this reason, there is a classification method known as Lauren’s criteria that has implications for therapy. The subtypes are as follows: intestinal, diffuse, and mixed types. Notably, diffuse-type adenocarcinoma tends to be more frequently diagnosed in female and younger patients, while the intestinal type is often associated with *Helicobacter pylori* infection and the presence of intestinal metaplasia [4,5,6]. Figure 1, which is from our database, shows the histological image of GC cells. Chemotherapy remains the main treatment option for metastatic GC; however, due to the tumor’s heterogeneity, the patients’ prognosis is poor. The emergence of molecular profiling of tumors in clinical settings paves the way for the application of novel targeted therapies with chemotherapy [4,5].

For instance, trastuzumab, a monoclonal antibody targeting human epidermal receptor 2 (HER2), was the first approved targeted therapy. Unfortunately, only 15–20% of patients can benefit from trastuzumab. As a second-line therapy, another targeted therapy, anti-vascular endothelial growth factor therapy was approved [4,5,6,7]. Current research is focused on improving GC treatment outcomes by identifying new therapeutic targets related to proteins involved in epithelial–mesenchymal transition (EMT) and cell–cell adhesion. EMT plays a vital role in cancer metastasis and is initiated by the breakdown of cell–cell adhesion structures such as tight junctions, adherens junctions, desmosomes, and gap junctions. Among these, claudins (CLDNs) exhibit high expression levels in certain cancers, including GC. The effectiveness of these approaches in an adjuvant setting is still being explored [8,9]. Aberrations in CLDN18.2, found predominantly in the genomically stable subgroup and diffuse histological subtype, represent promising targets for precision drugs like monoclonal antibodies. Zolbetuximab, a monoclonal antibody designed specifically for CLDN18.2, has exhibited effectiveness in phase II trials and, more recently, in the phase III SPOTLIGHT trial. The results demonstrated improved progression-free survival (PFS) and overall survival (OS) compared to standard treatment protocols. Moreover, the phase III GLOW trial demonstrated that zolbetuximab, when used in combination with chemotherapy, improved both PFS and OS in patients with CLDN18.2-positive, HER2-negative, locally advanced unresectable or metastatic GC [9,10,11]. Fibroblast growth factor receptor 2 (FGFR2), a member of the fibroblast growth factor family, is active in about 20–30% of GCs. This activation is primarily attributed to the amplification of the FGFR2 gene or the presence of splice variants. Bemarituzumab has the potential to target this activation in GC. To assess bemarituzumab’s effectiveness, the FIGHT study selected patients who showed FGFR2b membrane overexpression in at least 5% of tumor cells via immunohistochemistry (IHC) [4]. Additionally, there has been significant emphasis in the scientific community on using immune checkpoint inhibitors (ICIs) to prolong patient survival. The ICIs are specialized monoclonal antibodies that focus on disrupting the cytotoxic T-lymphocyte-associated antigen 4 (CTLA-4) and PD-1/PD-L1 pathway. The use of the anti-CTLA-4 inhibitor (Ipilimumab) was first established in the treatment of melanoma [12,13,14,15]. Currently, a wide spectrum of cancer types can be effectively addressed through anti-PD-1 agents (such as pembrolizumab and nivolumab), anti-PD-L1 inhibitors (like avelumab and atezolizumab), or through a co-administration of these agents with an anti-CTLA4 inhibitor [12,13,14,15,16].

Consequently, incorporating anti-programmed cell death protein 1 (PD-1) into first-line treatments for HER-2 negative GC improves clinical outcomes only in patients with high programmed death ligand 1 (PD-L1) expression. It should be emphasized that in the phase II randomized CheckMate 649 trial, it was found that the combination of nivolumab and chemotherapy is a recommended first-line treatment for patients with HER2-negative GC, especially those with a PD-L1 expression level of 5 or more, as assessed by the combined positive score (CPS) [11]. The Keynote-059 study (a phase II study) also showed that pembrolizumab plus chemotherapy demonstrates manageable safety and promising antitumor activity as first-line therapy in advanced GC, regardless of PD-L1 expression [12]. Additionally, the Keynote-811 study showed that pembrolizumab, in combination with trastuzumab and chemotherapy, significantly improved PFS in patients with metastatic HER2-positive GC, specifically with a PD-L1 CPS of 1 or more [14]. Moreover, patients who express high microsatellite instability seem to experience impressive clinical improvement and prolong survival. Despite the clinical application of immune checkpoint blockade, it appears to be beneficial only in a specific subgroup of patients, and we cannot actually predict in which patients this will be advantageous [13,14,15].

Furthermore, although there have been significant advances in diagnostic techniques and treatment methods, GC remains associated with a poor prognosis. Therefore, it is essential to investigate the immunomodulatory role of the tumor microenvironment (TME) and its influence on the effectiveness of ICIs. This is important crucial for the improvement of currently approved therapies [16].

The principal role of immune checkpoint blockade is to counteract the suppression of T cells caused by the tumor, effectively “reawakening” them. Consequently, the immune system is able to identify and eliminate the tumor through a series of orchestrated mechanisms [10,11,12,13,14].To be more specific, PD-1 is a 55-kDa transmembrane glycoprotein and contains an extracellular immunoglobulin variable (IgV)-like domain responsible for connecting with its ligands. Additionally, it has a cytoplasmic tail containing two tyrosine-based inhibitory motifs (ITIMs), the role of which is to suppress the immune system. Specifically, PD-1 and its ligands (PD-L1/PD-L2) are involved in the self-recognition and protection process, inhibiting the function of immune cells [14,15,16]. For this reason, a cancer cell may be recognized as a potential harm and targeted by the immune system; however, when interacting with the PD-1/PD-L1/PD-L2 signaling, it is recognized as part of the self and, therefore, rescued from the immune system targeting (cancer escape). The blockade of the PD-1 axis bypasses self-immuno-recognition and has been shown to be an effective treatment in several types of cancer. For this reason, many clinical trials have examined the benefit of PD-1/PD-L1 inhibitors in GC [12,13,17,18,19,20].

TME is a complex environment with diverse cell types and secreted factors playing a pivotal role in the progression of cancer. Tumor-associated macrophages (TAMs) represent a major component of this environment [21,22]. TAMs can facilitate cancer progression through different mechanisms, such as by accelerating tumor angiogenesis, promoting metastasis, and inactivating the adaptive immune system [22,23,24,25]. Additionally, TAMs are characterized by high plasticity and can be divided into a spectrum of different polarized types. This plasticity is defined by the disease stage, the affected tissue, and the host microbiota. The M1 and M2 polarized phenotypes represent the two edges of this spectrum. The M1 phenotype is the pro-inflammatory type, resulting in the killing of cancer cells, while the M2 phenotype is the anti-inflammatory type, promoting the evaluation of cancer cells [10,12,13,14]. Different markers are expressed by TAMs; two examples are CD164 and CD208, which are expressed by the M2 type but not by the M1 type. Both M1 and M2 types express the CD68 marker [3,26,27,28].

There is a significant increase in the proliferation of GC cells when in direct contact with M2 TAMs, compared to the indirect co-culture system. This suggests that, in addition to soluble factors derived from macrophages, direct cell-to-cell contact plays a role in fostering malignant characteristics in cancer cells [3,28,29]. Direct interaction between macrophages and cancer cells results in substantial differentiation of macrophages into the M2 phenotype. This change is characterized by the expression of CD163 and the secretion of IL-10. Furthermore, direct co-culture with M2 TAMs prompts signal transducer and activator of transcription 3 (STAT3) activation in cancer cells, contributing to increased cancer cell proliferation and progression [30,31,32,33,34].

It is believed that TAMs are regulated by signals belonging to two different categories: the “eat me” signals, which help TAMs function and the “do not eat me”, which inactivate them. The PD-1/PD-L1 axis is a member of the latter category. TAMs can express PD-1 on their surface and can form a ligand with PD-L1, which is expressed on the surface of cancer cells. M2 TAMs overexpress the PD-1 ligand and create an axis with the PD-L1 ligand of cancer cells. Therefore, an immunosuppressive environment is promoted, leading to aiding the cancer evaluation process and potentially facilitating cancer escape. However, there are many unanswered questions regarding the relationship between TAMs and the expression of the PD-1/PD-L1 axis [29,34,35,36,37,38].

The aim of this systematic review is to illustrate the correlation between the existence of TAMs and the expression of PD-1 in GC and to investigate the prognosis of patients in relation to the level of PD-1+ TAM expression.

## 2. Methods

### 2.1. Search Strategy

Our search strategy was deliberately broad in order to be comprehensive and to include all possible studies reporting the relationship between TAMs and the PD-1/PD-L1 axis. The algorithm we used included all the common synonyms related to GC, TAMs, and the PD-1/PD-L1 axis. For GC, we also used the following words: stomach, carcinoma, neoplasm, and tumor. Regarding TAMs, we included the following words: tumor-associated macrophages, tumor-infiltrating macrophages, and macrophages. Finally, for the PD-1/PD-L1 axis, we also added the PD-1 or, PD-L1 or PDL1 or PD1 or programmed cell death protein 1 or programmed death ligand 1. Furthermore, we applied our algorithm to three different databases (PubMed, Embase and Cochrane) to ensure a comprehensive retrieval of all the literature relevant to our study.

### 2.2. Study Inclusion and Exclusion Criteria

We included all articles that referred to studies with more than one patient with GC, investigating the correlation between the existence of TAMs and the cancer cells’ expression of PD-L1. Studies evaluating the prognosis of patients expressing PD-1+ TAMs were eligible for the review. On the other hand, we excluded case articles reporting case studies or examining non-human tissues. Studies where the full text was not available or was published in a language other than English were excluded.

### 2.3. Article Selection and Data Extraction

Initially, we only evaluated the titles and abstracts, discarding articles that did not meet our criteria. Following this, we thoroughly reviewed each of the remaining articles to determine their eligibility for our study (Figure 2). The extracted data were first reviewed for accuracy and then entered into an electronic database. This database recorded various details, such as the paper title, the author, the year of the publication, the gender and the mean and median age of the patients, the sample size, the histological type, the T-stage, the existence of infiltrated lymph nodes, the N-stage, the existence of metastasis, The American Joint Committee on Cancer (AJCC) staging 8th edition, the systematic therapy administered to patients, the type of the surgery, the follow-up of the patients and TAMs and PD-L1 cross tables.

## 3. Results

### 3.1. Characteristics of Included Studies

A total of ten articles met our inclusion criteria. The flow diagram, illustrating the final selection process for eligible articles, is shown in Figure 2. Seven studies investigated the correlation between the existence of TAMs and the expression of PD-L1 in GC cells [39,40,41,42,43,44,45], and three studies explored the prognosis of patients with GC expressing high levels of PD-1+ TAMs [46,47,48]. Due to the significant heterogeneity of the sample across the scrutinized studies, it was not possible to perform a cumulative statistical analysis of pooled available evidence. The articles considered for the first aim were published between 2017 and 2023, with a total number of patients of 1541 (Table 1). Only three articles published in 2018, 2020, and 2022, respectively, met our criteria for the second outcome.

Each study defined TAMs with different markers. The following markers used were: CD68, CD163, clever-1, CD68 and CD163, CD163 and CD208. In addition, PD-L1 is considered to be positive if it is >1%. The definitions of PD-L1, TAMs and PD-1+ TAMs are shown in Table 2 and Table 3. In Table 2 and Table 3, the methods of analysis used for the expression of PD-L1 in GC specimens are presented for each article. Specifically, after the initial processing of the slides, they were incubated with a primary antibody, PD-L1, with an anti-human PD-L1 monoclonal antibody [3,24,25,26,27,28,29,30,31,32,33,34,35,36,37,38,39,40,41,42,43,44,45,46,47,48]. In Ubukata et al.’s. study, PD-L1 expression was analyzed based on the staining of the cell membrane or cytoplasm of tumor cells, and the immunostaining score was calculated based on the number of positive cells and staining intensity. Samples with PD-L1 expression were categorized into three groups (no, low, and high expression) based on the Allred score [39]. In Ju et al.’s study, PD-L1 expression in tumor cells was graded into three groups depending on the staining (0, 1, 2—no, weak or moderate to intense staining) [40].

The expression of CD68 in the macrophage was scored as follows: 1 (20%), 2 (1–9%), 3 (10–20%), or 4 (>20%) [40]. In Ivanovic et al.’s study, the presence of PD-L1-positive tumor cells was categorized into four groups: negative (0), low (1–10%), moderate (11–49%) and strong (≥50%). Additionally, the PD-L1 status of GC was considered positive if ≥1% of tumor cells expressed PD-L1 [45].

Statistical analysis was performed, and Table 4 displays the methods used in each article and their *p*-value cut-off.

### 3.2. Correlation of TAMs and PD-L1

There is a correlation between the presence of TAMs and the expression of PD-L1. GC tissues with high PD-L1 expression show significantly greater macrophage infiltration. Conversely, low PD-L1 expression in GC is associated with reduced infiltration of TAMs. Whether these small differences in TAM expression may play a crucial role in immunotargeting/immunoescaping and whether they might influence immunotherapy outcomes remains a subject of research, as no prospective randomized clinical data are currently available [39,40,41,42,43,44,45].

Specifically, in Ju et al.’s study, it is reported that if there is no expression of PD-L1 in the cancer cells, the infiltration of CD68 cells is only 42%. On the other hand, if there is an expression of PD-L1, the infiltration of CD68 cells is estimated at 58%. Therefore, there is a positive correlation between the expression of PD-L1 and the infiltration of CD68 cells. Additionally, it is suggested that differentiated macrophages can induce PD-L1 expression in GC cells [40].

The study conducted by Junttila et al. revealed that 41% of the patients with GC had a high density of clever-1-positive macrophage, and 48% of them exhibited a high immune cell score. Additionally, it was noted that PD-L1 CPS (*p* = 0.474) was not statistically associated with survival. Similar results were observed between low and high densities of PD-1-positive lymphocytes (*p* = 0.204) and clever-1-positive macrophages (*p* = 0.428). Moreover, in the high ICS group, patients with high PD-L1, PD-1 or clever-1 had poor prognoses [41].

In addition, Huang et al.’s study shows that all TAMs express PD-L1. Notably, the macrophages characterized by the CD68+ CD206+ phenotype exhibit a substantially higher average expression of PD-L1 per patient when compared to other TAM subsets. The study also noted the heterogeneity of macrophages within the tumor, attributing it to the various markers used for their characterization [42].

Moreover, Harada et al.’s study shows that the density of CD68- and CD163-positive cells in PD-L1-positive GC samples was significantly higher than in PD-L1 negative GC samples (CD68 *p* = 0.0002; CD163 *p* < 0.0001). The study concludes that M2 macrophage infiltration could serve as a predictive marker for PD-L1 expression, thereby making M2 macrophages a potential therapeutic target in GC [43].

In Zhang et al.’s study, it was shown that PD-L1-positive cells were significantly increased in samples with high IL-10+ TAM infiltration. Given the established role of PD-L1/PD-1 interaction in stimulating macrophages’ IL-10 production, the outcome of this study suggests that PD-L1 could potentially play a role in facilitating the induction of IL-10-positive TAMs in GC. Nevertheless, further research is required to validate this [44].

Finally, Ivanovi et al.’s study identified a link between the presence of PD-L1 expression in tumor cells and the infiltration of macrophages. To be more precise, the proportion of macrophages within PD-L1-positive GC instances was found to be 2.6 times greater compared to the levels observed in PD-L1 negative cases [45].

Ubukata et al.’s study is the only study that did not find a statistically significant correlation between PD-L1 expression and CD163 (*p*-value = 0.6) [39].

It is believed that TAMs are recruited from both tissue-specific embryonic and monocytic-derived resident macrophages, with the latter source contributing to the growing body of TAMs. Macrophages can be differentiated by elevated levels of different types of cytokines, chemokines, growth factors and other signals from tumor and stromal cells. M1 macrophages secrete proinflammatory cytokines such as IL-12, tumor necrosis factor (TNF-α), chemokine ligand 1 (CXCL-10), and interferon (IFN)-γ and produce high levels of nitric oxide synthase (NOS), which is responsible for the metabolizing arginine to the nitric oxide. On the other hand, M2 macrophages secrete anti-inflammatory cytokines such as IL-4, IL-10, and IL-13 and express in high degrees arginase-1, CD206, and CD163 [33,39,40,41,42,43,44,45].

It seems that IL-6 and TNF-α are responsible for the high expression of PD-L1. The NF-κβ and STAT3 signaling pathways regulate macrophage-induced PD-L1 expression. STAT3 inhibitor C188-9 and IKK inhibitor BAY11-7821 can suppress the expression of PD-L1 [21]. In addition, it seems that TNF-α, p-65, and STAT3 have a good prognostic value in GC, and macrophages can promote the proliferation of GC cells by inducing the expression of PD-L1 [33,40].

In addition, the infiltration of TAMs is closely associated with tumor invasion and the development of metastasis. In this context, tumor cells release colony-stimulating factor 1, while TAMs release epithelial growth factor (EGF). These factors collectively promote the co-migration and invasion of both cell types toward blood vessels. It has been observed that macrophages from distant metastatic sites enhance the rate of invasion when compared to non-metastatic cells. Notably, the migration rate experiences a significant boost when a cell line containing macrophages is cultured under hypoxic conditions. In response to hypoxia, there is an upregulation in the expression of genes encoding disintegrin and metalloproteinase domain-containing protein 8 and 9 (ADAM8 and ADAM9), while the expression of matrix metallopeptidase 9 (MMP9) and tissue inhibitor of metalloproteinase 3 (TIMP3) decreases. This modulation in the expression of ADAMs, TIMP3, and MMP9 suggests that these genes may collectively contribute to the role of TAMs in facilitating the rapid and aggressive invasion characteristic of GC [3].

### 3.3. Prognosis of PD-1+ TAMs

Wang et al.’s study found that the macrophages in tumor tissues express a higher level of PD-1 compared to those in non-tumor tissues and in blood samples. PD-1+ TAMs were also accumulated in GC tissue [32,33,34]. In Kono et al. ’s study, the correlation between the presence of PD-1+ TAMs and clinicopathological variables is examined. In this study, high PD-1+ TAMs is defined as the frequency of PD-1+ macrophages: ≥ 0.85%. Specifically, it seems that the presence of PD-1+ TAMs is significantly higher in patients with lymph node metastasis and at age 75 or more. The presence of PD-1+ TAMs in patients aged 75 or more is 3.31%, but the presence of PD-1+ TAMs in patients aged lower than 75 is 1.42%. Furthermore, the percentage of TAMs in patients with lymph node metastasis and in those without lymph node metastasis is 3.83% and 0.54%, respectively [46].

Available studies indicate that the prognosis of GC patients is correlated with the expression of the PD-1+ TAM axis.

In GC, the expression of PD-1+ TAMs seems to be associated with disease progression and early recurrence in those patients. In Kono et al.’s study, the five-year disease-specific survival rates among patients with GC cells expressing high PD-1+ TAMs and low PD-1+ TAMs are 65.8% and 85.9%, respectively. The comparison of the phenotypic features of GC-infiltrating PD-1+ TAMs revealed that PD-1+ TAMs are polarized to the M2-like type, expressing higher levels of M2 markers such as IL-10 and CCL1. It was also shown that the phagocytotic ability of PD-1+ TAMS is lower than PD-1- TAMs [44]. Overall, it seems that PD-1+ TAMs have a pro-tumorigenic activity because they reduce the proliferation of CD8 T cells. On the other hand, PD-1+ TAMs do not correlate with the activity of CD4 T cells [47,48].

Furthermore, Wei et al. observed that TAMs within the TME exhibited a high degree of PD-L1 expression. Additionally, more infiltration of macrophages was associated with poor prognosis [48]. Moreover, a strong correlation was identified between increased macrophage accumulation at the time of diagnosis and an unfavorable prognosis among patients with GC [46,47,48]. Specifically, a correlation was found between a higher accumulation of macrophages in tumors at the time of diagnosis and a poorer prognosis in GC patients. These results suggest that tumors tend to advance preferentially in the presence of a heightened macrophage infiltrate. Conversely, tumors exhibiting a favorable T-cell immunophenotype may experience partial restraint from the immune system, resulting in less frequent progression to advanced stages [49].

## 4. Discussion

Within the TME, there are various types of cells, including immune cells, that play a crucial role in cancer progression. TAMs are the most abundant population in the TME, and their role is controversial as they can either kill tumor cells or contribute to tumor growth. This dual role of macrophages is mainly classified into two different types: M1 and M2 macrophages. There is actually a spectrum of different phenotypes that macrophages can be polarized into, with M1 and M2 types representing the two extremes of this spectrum. For instance, M2 macrophages release epidermal growth factor or transforming growth factor-β and can stimulate the EMT [29,31,32,33].

Several studies have reported a correlation between the existence of TAMs and the expression of PD-L1 in cancer cells. In GC tissue samples, macrophage infiltration is associated with higher expression of PD-L1 in tumor cells. Conversely, the low expression of PD-L1 correlates with low TAM infiltration [38,39,40,41,42,43,44,45]. TAMs can also express PD-L1 on their surface and directly suppress T cells. Additionally, PD-L1 expression can be a predictive marker for patient outcomes after ICIs [39,40,41,42,43,44,45]. Nevertheless, the observed differences in TAM infiltration were small (numerically less than 10%), underscoring that the PD-L1/TAMs interaction’s involvement in immunotargeting/immunoescaping is probably a qualitative rather than a quantitate process [39,40,41,42,43,44,45,46,47,48]. In addition, TAMs can express PD-1+ on their surface and PD-1+ TAMs exist in higher concentrations in GC cells than in non-cancerous cells. Specifically, two categories of signals regulate the function of TAMs. One category is the “eat me” signals that activate the phagocytosis of macrophages. On the other hand, the other category is the “do not eat me” signals where PD-1 belongs. The role of the “do not eat me” signals is to inactivate the phagocytosis of macrophages [27,28]. PD-1 is a 55-kDa transmembrane protein, and it belongs to the B7-CD28 superfamily. There are two known ligands, PD-L1 and PD-L2. PD-1 has an extracellular domain responsible for conjunction with its ligands, along with a cytoplasmic tail containing an ITIM domain (immunoreceptor ITIM) with two tyrosine bases. When the PD-1/PD-L1 axis is complete, phosphorylation occurs in these tyrosine bases within the ITIM domain. This phosphorylation mediates the recruitment and activation of two proteins, SHP1 and SHP2. Consequently, there is a dephosphorylation of myosin IIA, ultimately leading to the inhibition of cytoskeleton rearrangement and the inactivation of macrophages. Therefore, the PD-1/PD-L1 interaction plays a crucial role in the regulation of TAMs because it is associated with their dysfunction [27,28,35].

Furthermore, the prognosis of those patients correlates with the presence of PD-1+ TAMs. The expression of PD-1+ TAMs is associated with a poor prognosis and early recurrence. Consequently, therapies targeting the PD-1 pathway may be less effective in GC. Therefore, in lieu of these methods, therapies that target the PD-1 pathway combined with anti-TAMs directed treatment should be investigated in a GC setting [47,48,49].

The use of TAMs as prognostic and diagnostic biomarkers holds great promise. In fact, TAMs have already demonstrated significant potential as diagnostic biomarkers in other types of cancer, including multiple myeloma, breast cancer, prostate cancer, and pancreatic cancer. Moreover, TAMs show promise as prognostic markers, providing valuable insights into outcomes for patients with lung cancer, esophageal squamous cell carcinoma, and bladder cancer [3,39,40,41,42,43,44,45,46].

Studies have revealed that the quantity of TAMs within the tumor stroma can predict important factors such as tumor size, stage, and the likelihood of metastasis in GC. This valuable information enhances the accuracy of prognosis and aids in tailoring individualized treatment plans for patients with GC. Notably, patients with higher levels of TAMs tend to exhibit shorter overall survival rates compared to those with lower levels of TAMs. Therefore, TAMs may serve as a practical tool for identifying at-risk patients, facilitating early diagnosis, and forming more precise prognostic assessments [3,39,40,41,42,43,44,45,46].

Nowadays, therapeutic strategies target TAM infiltration. Generally, TAM targeting approaches aim to impede TAM infiltration in tumors, achieved either by inhibiting recruitment or inducing direct elimination. Additionally, there are approaches focusing on reshaping their pro-tumorigenic polarization to stimulate their anti-tumor capabilities. Consequently, there are few ongoing phase I or II clinical trials. It is crucial to gain a better understanding of the role of TAMs and their correlation with PD-1/PD-L1 for the development of effective treatments [50,51].

The initial aim of this review was to conduct a statistical meta-analysis of the available results. Unfortunately, due to the heterogeneity of the articles and the lack of accurate data pertaining to our question, such an analysis was not feasible. Therefore, it is imperative that new research be conducted with a precise comparison of the expression of PD-L1, the infiltration of TAMs, and the prognosis of PD-1+ TAMs.

## 5. Conclusions

To conclude, TAMs play a pivotal role in the regulation of PD-1/PD-L1 networking and the progression of GC cells. Unfortunately, the data of the available studies are scarce, sparse, and inadequate for cumulative statistical pooled analyses of the available evidence. Further studies are thereafter required to investigate PD-1+ TAMs axes expression and their impact on GC survival. The correlation of TAMs and PD-1/PD-L1 expression looks very promising for new therapies in this setting. To validate this, more in-patient studies must be carried out [39,40,41,42,43,44,45,46,47,48].

## Figures and Tables

**Figure 1 cancers-16-00196-f001:**
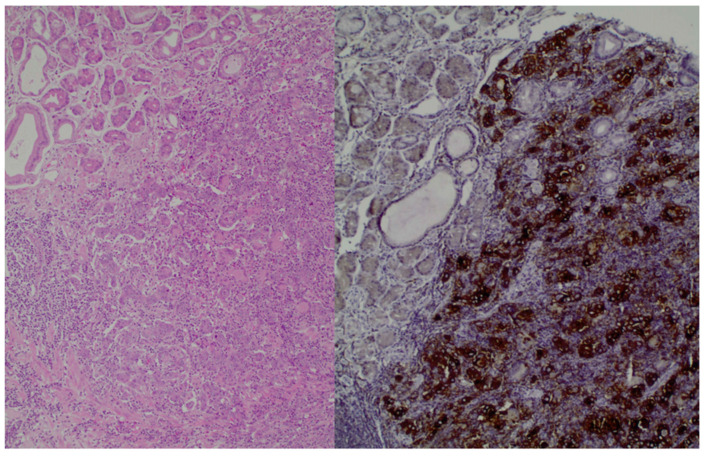
(**left**): Gastric adenocarcinoma, Hematoxylin and eosin (H+E) × 200, (**right**): Gastric adenocarcinoma, PD-L1 immunostaining (DAKO), DAB × 200.

**Figure 2 cancers-16-00196-f002:**
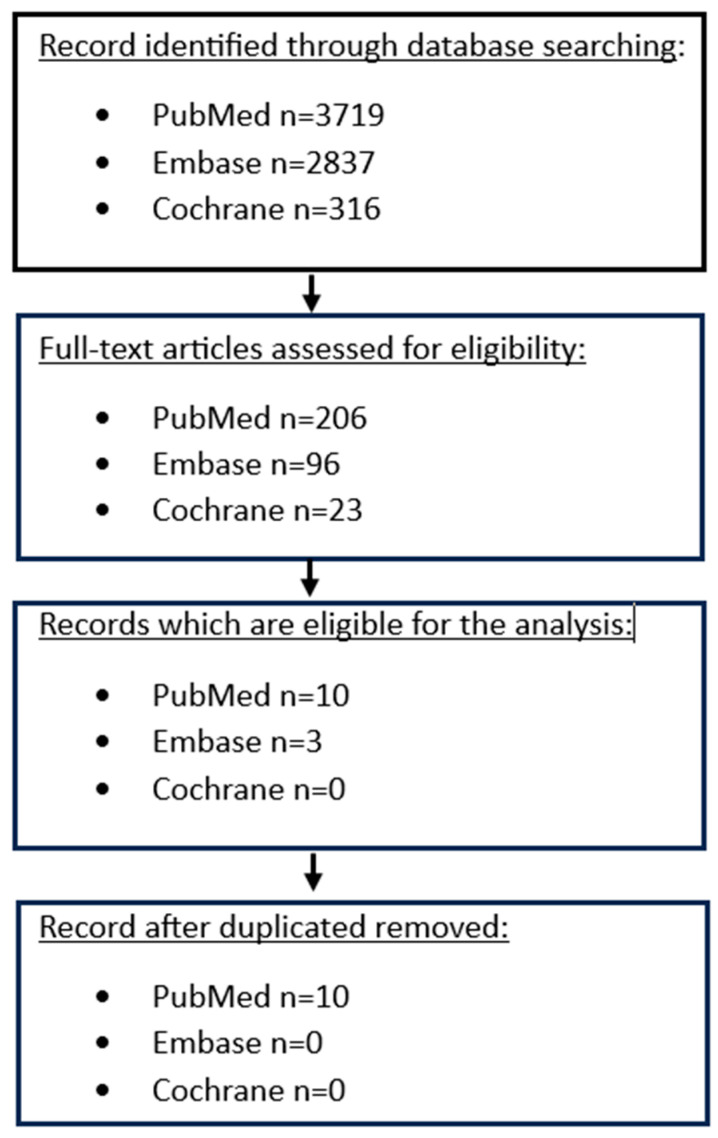
Flowchart depicting the search strategy followed in our analysis.

**Table 1 cancers-16-00196-t001:** Characteristics of the included studies.

	Author	Gender n(%)		Sample Size	Used in the Analysis	T-Stage				Lymphnodes		N-Stage					Metastasis	AJCC Staging 8th Edition				Surgery	
		Male	Female			T1	T2	T3	T4	YES	NO	N0	N1	N2	N3	YES	NO	I	II	III	IV	YES	NO
**Correlation of TAMS and PD-L1**																							
1	**Ubukata Y (2020)** [39]	76 (71)	31 (29)	635	107	107	0	0	0	45	62	NA	NA	NA	NA	NA	NA	NA	NA	NA	NA	107	0
2	**Ju X (2020)** [40]	NA	NA	95	95	NA	NA	NA	NA	NA	NA	NA	NA	NA	NA	NA	NA	NA	NA	NA	NA	NA	NA
3	**Junttila A (2020)** [41]	61 (51.7)	57 (48.3)	132	118	34	52	32	4	40	78	40	53	18	7	4	114	32	51	31	4	118	0
4	**Huang Y.K (2019)** [42]	36 (64.3)	20 (35.7)	56	56	3	13	39	1	33	23	23	18	10	5	3	53	NA	NA	NA	NA	NA	NA
5	**Harada K (2017)** [43]	134 (62)	82 (38)	217	216	31	104	66	14	144	67	67	84	27	33	59	154	67	49	38	59	216	0
6	**Zhang H (2022)** [44]	575 (67.5)	270 (32.5)	932	852	110	140	261	334	550	302	302	152	162	230	0	852	169	224	448	0	852	0
7	**Ivanovic T (2023)** [45]	66 (68)	31 (32)	97	97	NA	NA	NA	NA	NA	NA	NA	NA	NA	NA	3	94	14	31	49	3	97	0
				2164	1541																		
**Prognosis of PD-1+ TAMS**																							
1	**Kono Y (2020)** [46]	75 (73.5)	27 (26.5)	102	102	12	NA	NA	NA	48	54	NA	NA	NA	NA	85	17	NA	NA	NA	0	102	0
2	**Wang F (2018)** [47]	17 (65.4)	9 (34.6)	26	26	NA	NA	NA	NA	NA	NA	NA	NA	NA	NA	NA	NA	5	9	9	3	NA	NA
3	**Wei Y (2022)** [48]	8 (80)	2 (20)	10	10	NA	NA	NA	NA	NA	NA	NA	NA	NA	NA	1	9	NA	NA	6	2	10	0
				138	138																		

**Table 2 cancers-16-00196-t002:** Definition of PD-L1 and TAMs.

Correlation of TAMS and PD-L1				
	Author	Definition of PD-L1	Definition of TAMS	Method of Analysis
**1**	**Ubukata Y (2020)** [39]	No/low and high expression	CD163 level was determined	Immunohistochemistry
**2**	**Ju X (2020)** [40]	0 (no staining), 1 (weak staining), 2 (moderate to intense staining).	CD68 in macrophage was scored as 1 (<1%), 2 (1–9%), 3, or 4 (>20%)	Immunohistochemistry
**3**	**Junttila A (2020)** [41]	PD-L1 CPS was ≤1% or >1%/Divided into two groups	Clever-1 was defined to be high when median ≥ 15 in macrophages	Immunohistochemistry
**4**	**Huang Y.K (2019)** [42]	Low or high expression	Macrophages were CD68 positive/M1-like TAM populations were identified based on the absence of CD163 and CD206/M2-like TAM populations were identified by the presence of CD163 and/or CD206	Immunohistochemistry
**5**	**Harada K (2017)** [43]	PD-L1 positive if >1% of tumor cells expressed PD-L1	The density of CD68- and CD163-positive cells was evaluated	Immunohistochemistry
**6**	**Zhang H (2022)** [44]	Positive or negative PD-L1 expression	For the CD68+ macrophages density, ≤66/HPF (at ×200 magnification) was defined as low and ≥67/HPF was defined as high/For the IL-10+ CD68+ cells density, ≤9/HPF was defined as low and ≥10/HPF was defined as high.	Immunohistochemistry
**7**	**Ivanovic T (2023)** [45]	PD-L1-positive tumor cells estimated as negative (0), low	The density of macrophage infiltration was valued as 0 (no infiltration), 1 (sparse), 2 (peripheral) and 3 (dense and intermingled with tumor cells)	Immunohistochemistry

**Table 3 cancers-16-00196-t003:** Definition of PD-1+ TAMs.

Prognosis of PD-1+ TAMs			
Author		Definition of PD1 + Macrophages	Method of Analysis
**1**	Kono Y (2020) [46]	The frequency of PD-1+ macrophages was represented by the ratio of the number of PD-1+ CD68+ cells to that of CD68+ cells.	Immunohistochemistry
**2**	Wang F (2018) [47]	PD+ or PD- maxrophages	Immunohistochemistry
**3**	Wei Y (2022) [48]	High or low expression of PD-L1 in macrophages/Tumor macrophages expressed higher levels of CD68 and PPARg and lower levels of HLA-DR.	Mass cytometry by time of flight (CyTOF) combined with genomic bioinformatic analysis

**Table 4 cancers-16-00196-t004:** Statistical methods were used and a cut-off *p*-value.

Correlation of TAMS and PD-L1			
	Author	Statistical Methods Were Used	Statistically Significant Cutoff
**1**	**Ubukata Y (2020)** [39]	Student’s *t*-test and the χ^2^ test	*p* values < 0.05
**2**	**Ju X (2020)** [40]	Kaplan-Meier methods, Log-Rank test and Student *t*-test	*p* values < 0.05
**3**	**Junttila A (2020)** [41]	Kaplan-Meier method, Log-Rank, Chi-square test and Univariate and multivariate Cox proportional hazards regression models	*p* values < 0.05
**4**	**Huang Y.K (2019)** [42]	Mann–Whitney U test (two-tailed), Spearman and Pearson correlation, Chi-square analysis and Kaplan–Meier analysis (Log-rank, Mantel-Cox test)	*p* values < 0.05
**5**	**Harada K (2017)** [43]	Kruskal Wallis test, Pearson chi-square test, Fisher’s exact and Cox regression model test	*p* values < 0.05
**6**	**Zhang H (2022)** [44]	Chi-squared test, Student *t*-test, Kaplan–Meier curves, Cox proportional hazards regression model and Spearman correlation analysis	*p* values < 0.05
**7**	**Ivanovic T (2023)** [45]	*t*-test, Mann–Whitney U test, χ^2^ test, Pearson correlation coefficient, Kaplan–Meier curve and Log-rank test	*p* values < 0.05
**Prognosis of PD-1+ TAMs**			
**1**	**Kono Y (2020)** [46]	Paired *t*-test, Mann-Whitney U test, Kaplan-Meier method, Log-rank test and Cox’s proportional hazards model	*p* values < 0.05
**2**	**Wang F (2018)** [47]	Bonferroni post test and Pearson’s correlation analysis	*p* values < 0.05
**3**	**Wei Y (2022)** [48]	Paired sample *t*-test	*p* values < 0.05

## Data Availability

The data presented in this study are available in this article.

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
