# Peer review of "TAMs and PD-1 Networking in Gastric Cancer: A Review of the Literature"

_cancers, 2023, doi:10.3390/cancers16010196_

Round 1
Reviewer 1 Report
Comments and Suggestions for Authors
This manuscript by Yerolatsite et al is a systematic review of published literature that focuses on describing the correlation between the presence of tumor-associated macrophages (TAMs) and PD-L1 expression in gastric cancer. Through a search of three databases, the investigators identified ten relevant articles investigating human patients that described PD-L1 positivity in gastric cancer and TAMs. The authors identified nine studies that identified PD-L1 and TAM correlation, as well as three studies which looked at clinical features or prognosis. TAMs seem to play an important role in immune regulation across multiple tumor types. The data thus far is scarce for gastric cancer, and the investigators are able to capture relevant studies. Overall, this is an important topic that should be investigated in relationship to the gastric cancer TME. The study can improve on the organization and English writing. As immunotherapy with PD-1 inhibitors is approved in patients with metastatic gastric cancer, the authors should describe what is known in terms of activity to better contextualize why investigation of the gastric cancer TME and TAMs specifically is important. Since there are only a few studies available, there should be more specific description of details of these studies if available as detailed below. The discussion could also be strengthened by describing current or emerging therapies to target TAMs.
Abstract: There are grammatical errors and typos. For example, “Since TAMs can express PD-1+ [should remove +] in their surface and can make a ligand with PD-L1” should be rephrased. There should be a space between 2277 and patients, PDL-1 should be PD-L1 (should be consistent throughout manuscript).
Line 55- “systematic” should be “systemic”
Would consider rephrasing (Line 55 – 58) as there have been improvements in therapies including new targeted therapies such as HER2 and Claudin 18 that have improved survival.
Would be consistent- sometimes PD-L1 is spelled PDL-1
I am uncertain what is meant by “strategic amalgamation” in line 68
Typo immunostaingN in Figure 1 title
Line 57-58: ICI is abbreviation for immune checkpoint inhibitors earlier, but her it is abbreviation for immune checkpoint blockade. This should be corrected
The authors state that many clinical trials have examined the benefit of PD-1/PD-L1 inhibitors in gastric cancer (Lines 81-82). They cite some review articles which don’t seem to be that relevant References 9-11 (i.e., Feng et al which is about phagocytosis checkpoints, Gordon et al which is on PD-1 expression in TAMs). It seems these references are misplaced. It would be helpful to cite some of these pivotal trials investigating anti-PD1 therapy in gastric cancer (i.e., KEYNOTE-059, CheckMate-649 to name a few). It would also be important to note that anti-PD1 therapy by itself has only modest benefit in gastric cancer (i.e., in KEYNOTE-059, the objective response rate with pembrolizumab was 11.6%). Also, ipilimumab plus nivolumab did not improve survival in CheckMate-649. I think it would be important to include some perspective of the known activity of PD1 therapy, and why investigating roles for other immune cells like TAMs that may play an immunomodulatory role in the TME would be important in the context of currently approved therapies.
Line 94- it is unclear what is meant by M2 TAMs promoting the evaluation of cancer cells
Line 103- This sentence is incomplete
Line 144- They reference a flow diagram in Figure 3; however, this appears to be Figure 2 that they should be referencing.
Line 153- It is not clear how the TAM populations are defined and what combinations of biomarkers are being used. For instance, it states that “CD163 and CD163 and CD208.” Should this read CD163 and CD208?
PD-L1 can be tested by several different antibodies and different scoring systems (such as combined positive score [CPS] and total positive score [TPS]). Are these studies utilizing the same PD-L1 antibody? Clinically, PD-L1 combined positive score (not percentage) is often used, and takes into account both PD-L1 expression on tumor cells and immune cells. Are all of the studies utilizing PD-L1 on only tumor cells?
There are typos in Table 1- i.e, lymph nodes is spelled incorrectly
Line 159- How is low expression of PD-L1 being defined here?
The sentence describing factors promoting macrophage differentiation should be reworded, as it makes it seem like HLA-Dra and CD209 are cytokines
Abbreviations should be defined at the first instance (ie TNFa in 180 is used previously)
Again, it should be explained how high PD-1+ and low PD-1+ macrophages are defined (Line 219)
In the description of Wei et al, can the authors provide more details on correlation of macrophage accumulation and the effect on prognosis (i.e., is this looking at overall survival or progression-free survival, what is the magnitude of effect).
Some of the discussion is repetitive in the discussion and results, i.e. discussion of cytokines and other factors that regulate TAMs. The authors may consider consolidating this in the discussion.
There are typos in the references (i.e., Small is spelled incorrectly in reference 29)
Graphic/table to summarize the key findings on correlation would be helpful although given the heterogeneity of the studies, this may be difficult.
Comments on the Quality of English LanguageThere are several typos and grammatical errors that affect the readability of the manuscript.
Author Response
Please see attached word document for the response.

Reviewer 2 Report
Comments and Suggestions for Authors
cancers-2700720
Type of manuscript: Review
Title: TAMs and PD-1 networking in gastric cancer: a review of the literature
Authors: Melina Yerolatsite *, Nanteznta Torounidou, Aristeidis Gogadis, Fani Kapoulitsa, Panagiotis Ntellas, Evangeli Lampri, Maria Tolia, Anna Batistatou, Konstantinos H. Katsanos, Davide Mauri
In general, the current review is well-written to match the title and supports the main conclusions of this review paper. However, there is significant room for improvement, and numerous fundamental errors in basic paper writing, as pointed out below, have been identified. Please consider these comments for further revisions to the paper.
Specific comments are as follows:
[Major concern]
1. While it's true that there aren't many research papers that encompass TAMs, PD-1, and gastric cancer altogether, there are, as you are well aware, a substantial number of papers on TAMs, PD-1, and gastric cancer individually. It is evident that there is a need for more in-depth review papers on these topics.
2. This review paper contains an abstract. However, the addition of a "simple summary" has raised questions about its necessity.
3. Abbreviations: Most journals require that an abbreviation be spelled out at its first occurrence in the text, followed by the abbreviation in parentheses. (Exception: If the abbreviation is on the journal's list of permitted abbreviations, this need not be done.) Thereafter, only the abbreviation may be used. Note also that abbreviations need to be independently defined in the abstract and the main text of the paper. Abbreviations need not be introduced if they are not used again.
Some abbreviations have been mentioned earlier, but they are being repeated here. Therefore, it would be beneficial to systematically review and clarify the abbreviation usage from the beginning.
4. English: The English composition of the paper is generally well-done. However, some names of disease or compound names are written in uppercase letters even though they are not the first letter of the sentence or proper nouns. Please make corrections throughout the text and in the figures.
5. When writing a paper, it is crucial to accurately represent key terms and scientific terminology. In this paper, there are some critical keywords that are written in a confusing manner. For instance, "PD-L1" is conventionally spelled as such, but it is written as "PDL-1." Furthermore, the paper alternates between "PD-1/PD-L1 axis", “PD-1.PD-L1 axis”, and "PD-1-PD-L1." These inconsistencies are found in several places, so please make sure to identify and consistently use the correct spellings throughout the paper.
6. Citations and references in the text: In the text, reference numbers should be placed before the punctuation; for example [1], [1–3] or [1,3]. For embedded citations in the text with pagination, use both parentheses and brackets to indicate the reference number and page numbers; for example [5] (p. 10). or [6] (pp. 101–105).
7. Reference section: Author should consult and peruse carefully recent issues of the journal, Cancers, for format and style. Also double-check the abbreviations of journal names.
8. The paper is labeled as a review paper; however, it only cites a total of 29 references. Typically, when writing a review paper on a specific topic, one would expect around 100 references or so. In other words, it is a common practice to draw from a wide range of sources when composing a review paper. This review paper, on the other hand, is exceptional in that it cites a very limited number of references. Nevertheless, upon reviewing the content, there is ample room to include more references.
9. Despite being labeled as a review paper, Figure 1 is included in the manuscript. However, it appears that Figure 1 may not be necessary for the content. Furthermore, in the current Figure 1, labels 'a' and 'b' are missing. In conclusion, given that this is a review paper, I recommend omitting the figure altogether.
[Minor concern]
1. Line 20: ‘Tumor Microenvironment’ should be re-written as ‘Tumor microenvironment’.
2. Lines 22 and 23: Define PD-1 and PD-L1 in the abstract.
3. Line 33: Define GC in the abstract.
4. Lines 39 and 41: After this sentence, add reference(s).
5. Line 53: Define H. pylori and these words should be written in italics.
6. Line 57: ‘Immune’ should be written as ‘immune’.
7. Line 60: Define H+E.
8. Line 73: Define IgV.
9. Line 91 and more: Define M1 and M2.
10. Lines 99, 260, 261, and 262: “eat me” and “don’t et me: Please ensure that these phrases should be written all consistently.
11. Line 136: Define AJCC.
12. Line 162: ‘Ju X.’s’ is not a proper citation.
13. Lines 178 and more: TNF-a should be written as TNF-α.
14. Line 183: IL-10, IL-13, and IL-4: Generally, in cases like this, unless there is a specific reason, list them in numerical order.
15. Line 251: NF-Kβ should be written as NF-κB.
Overall, the manuscript can be rejected and encourage re-submission.
Comments on the Quality of English Languagecancers-2700720
Type of manuscript: Review
Title: TAMs and PD-1 networking in gastric cancer: a review of the literature
Authors: Melina Yerolatsite *, Nanteznta Torounidou, Aristeidis Gogadis, Fani Kapoulitsa, Panagiotis Ntellas, Evangeli Lampri, Maria Tolia, Anna Batistatou, Konstantinos H. Katsanos, Davide Mauri
In general, the current review is well-written to match the title and supports the main conclusions of this review paper. However, there is significant room for improvement, and numerous fundamental errors in basic paper writing, as pointed out below, have been identified. Please consider these comments for further revisions to the paper.
Specific comments are as follows:
[Major concern]
1. While it's true that there aren't many research papers that encompass TAMs, PD-1, and gastric cancer altogether, there are, as you are well aware, a substantial number of papers on TAMs, PD-1, and gastric cancer individually. It is evident that there is a need for more in-depth review papers on these topics.
2. This review paper contains an abstract. However, the addition of a "simple summary" has raised questions about its necessity.
3. Abbreviations: Most journals require that an abbreviation be spelled out at its first occurrence in the text, followed by the abbreviation in parentheses. (Exception: If the abbreviation is on the journal's list of permitted abbreviations, this need not be done.) Thereafter, only the abbreviation may be used. Note also that abbreviations need to be independently defined in the abstract and the main text of the paper. Abbreviations need not be introduced if they are not used again.
Some abbreviations have been mentioned earlier, but they are being repeated here. Therefore, it would be beneficial to systematically review and clarify the abbreviation usage from the beginning.
4. English: The English composition of the paper is generally well-done. However, some names of disease or compound names are written in uppercase letters even though they are not the first letter of the sentence or proper nouns. Please make corrections throughout the text and in the figures.
5. When writing a paper, it is crucial to accurately represent key terms and scientific terminology. In this paper, there are some critical keywords that are written in a confusing manner. For instance, "PD-L1" is conventionally spelled as such, but it is written as "PDL-1." Furthermore, the paper alternates between "PD-1/PD-L1 axis", “PD-1.PD-L1 axis”, and "PD-1-PD-L1." These inconsistencies are found in several places, so please make sure to identify and consistently use the correct spellings throughout the paper.
6. Citations and references in the text: In the text, reference numbers should be placed before the punctuation; for example [1], [1–3] or [1,3]. For embedded citations in the text with pagination, use both parentheses and brackets to indicate the reference number and page numbers; for example [5] (p. 10). or [6] (pp. 101–105).
7. Reference section: Author should consult and peruse carefully recent issues of the journal, Cancers, for format and style. Also double-check the abbreviations of journal names.
8. The paper is labeled as a review paper; however, it only cites a total of 29 references. Typically, when writing a review paper on a specific topic, one would expect around 100 references or so. In other words, it is a common practice to draw from a wide range of sources when composing a review paper. This review paper, on the other hand, is exceptional in that it cites a very limited number of references. Nevertheless, upon reviewing the content, there is ample room to include more references.
9. Despite being labeled as a review paper, Figure 1 is included in the manuscript. However, it appears that Figure 1 may not be necessary for the content. Furthermore, in the current Figure 1, labels 'a' and 'b' are missing. In conclusion, given that this is a review paper, I recommend omitting the figure altogether.
[Minor concern]
1. Line 20: ‘Tumor Microenvironment’ should be re-written as ‘Tumor microenvironment’.
2. Lines 22 and 23: Define PD-1 and PD-L1 in the abstract.
3. Line 33: Define GC in the abstract.
4. Lines 39 and 41: After this sentence, add reference(s).
5. Line 53: Define H. pylori and these words should be written in italics.
6. Line 57: ‘Immune’ should be written as ‘immune’.
7. Line 60: Define H+E.
8. Line 73: Define IgV.
9. Line 91 and more: Define M1 and M2.
10. Lines 99, 260, 261, and 262: “eat me” and “don’t et me: Please ensure that these phrases should be written all consistently.
11. Line 136: Define AJCC.
12. Line 162: ‘Ju X.’s’ is not a proper citation.
13. Lines 178 and more: TNF-a should be written as TNF-α.
14. Line 183: IL-10, IL-13, and IL-4: Generally, in cases like this, unless there is a specific reason, list them in numerical order.
15. Line 251: NF-Kβ should be written as NF-κB.
Overall, the manuscript can be rejected and encourage re-submission.
Author Response

(The authors gave the same response as above.)

Reviewer 3 Report
Comments and Suggestions for Authors
The authors aimed to systematically review published data using MEDLINE, PubMed, Embase, and Cochrane databases to analyze articles investigating the coexistence of TAM and PD-1 in gastric cancer and the prognosis of patients expressing high levels of PD-1+ TAM. However, apart from the selection of papers, no standardization or statistical methods were employed to analyze the collection of selected article. Only individual paper descriptions are provided, failing to yield a systematic analysis of the already published papers. Consequently, the review remains a general overview of existing articles and does not present any new findings from systematic analysis. In addition, some of the papers included in Table 1 appear to be poorly selected and poorly described.
1. The manuscript lacks tables or figures to visually represent the analysis. Authors could include tables categorizing the types of markers for TAM, the expression analysis methods used (histochemistry, RNA-seq, etc.), as well as the statistical methods employed, along with the results for each article.
2. line 160, In Ubukata's study, p value is 0.6. It is non-significant difference. The data in this ref did not support the correlated expression between PD-L1 and CD163 (TAM maker)
3. line 162. The author's description does not match the data in the paper although this papers clearly showed the correlation of CD68 and PD-L1.
4. The article no 3 and 5 of correlation of TAM and PDL-1 in Table1 : No data or results were found demonstrating a correlation between TAM and any macrophage markers.
Comments on the Quality of English LanguageThere are a few editorial errors, but it is written in easy-to-understand English.
Author Response
Please see attached word document for the response..

Round 2
Reviewer 2 Report
Comments and Suggestions for Authors
cancers-2700720-v2
Type of manuscript: Review
Title: TAMs and PD-1 networking in gastric cancer: a review of the literature
Authors: Melina Yerolatsite *, Nanteznta Torounidou, Aristeidis Gogadis, Fani Kapoulitsa, Panagiotis Ntellas, Evangeli Lampri, Maria Tolia, Anna Batistatou, Konstantinos H. Katsanos, Davide Mauri
The revised manuscript has been greatly improved and has been very helpful for reading and understanding. However, following issues need to be considered prior to considering the manuscript of publication. However, despite being pointed out in the first review and specifically instructed for corrections, I am dismayed to discover that several areas in the resubmitted paper have not been corrected. I strongly urge you to rectify these issues consistently from the beginning once again
Specific comments are as follows:
[Major concern]
1. When writing a paper, it is crucial to accurately represent key terms and scientific terminology. In this paper, there are some critical keywords that are written in a confusing manner. For instance, "PD-L1" is conventionally spelled as such, but it is written as "PDL-1." Furthermore, the paper alternates between "PD-1/PD-L1 axis", “PD-1.PD-L1 axis”, and "PD-1-PD-L1." These inconsistencies are found in several places, so please make sure to identify and consistently use the correct spellings throughout the paper. Examples: ‘PD-1-PD-L1’ is still there at Lines 391, 424, and 428. In my opinion, it seems appropriate to consistently refer to these as PD-1/PD-L1 as well.
[Minor concern]
1. Line 12: ‘the Programmed cell death protein 1(PD-1)/Programmed death-ligand 1 (PD-L1) pathway’ should be re-written as ‘the programmed cell death protein (PD-1)/programmmed death-ligand 1 (PD-L1) pathway’. There is no need to format these in bold.
2. Line 56: ‘Helicobacter pylori’: no need to format these in bold.
3. Line 250: ‘Ju X.et al.’ should be written as ‘Ju et al.’. Ju is a last name.
4. Line 321: TNF-a should be written as TNF-α.
5. Line 337: ‘3.83% and 0.54% respectively’ should be written as ‘3.83% and 0.54%, respectively’.
6. Reference section: Author should consult and peruse carefully recent issues of the journal, Cancers, for format and style. Also double-check the abbreviations of journal names.
Overall, the manuscript can be considered to publication after minor revision as indicated above.
Comments on the Quality of English Languagecancers-2700720-v2
Type of manuscript: Review
Title: TAMs and PD-1 networking in gastric cancer: a review of the literature
Authors: Melina Yerolatsite *, Nanteznta Torounidou, Aristeidis Gogadis, Fani Kapoulitsa, Panagiotis Ntellas, Evangeli Lampri, Maria Tolia, Anna Batistatou, Konstantinos H. Katsanos, Davide Mauri
The revised manuscript has been greatly improved and has been very helpful for reading and understanding. However, following issues need to be considered prior to considering the manuscript of publication. However, despite being pointed out in the first review and specifically instructed for corrections, I am dismayed to discover that several areas in the resubmitted paper have not been corrected. I strongly urge you to rectify these issues consistently from the beginning once again
Specific comments are as follows:
[Major concern]
1. When writing a paper, it is crucial to accurately represent key terms and scientific terminology. In this paper, there are some critical keywords that are written in a confusing manner. For instance, "PD-L1" is conventionally spelled as such, but it is written as "PDL-1." Furthermore, the paper alternates between "PD-1/PD-L1 axis", “PD-1.PD-L1 axis”, and "PD-1-PD-L1." These inconsistencies are found in several places, so please make sure to identify and consistently use the correct spellings throughout the paper. Examples: ‘PD-1-PD-L1’ is still there at Lines 391, 424, and 428. In my opinion, it seems appropriate to consistently refer to these as PD-1/PD-L1 as well.
[Minor concern]
1. Line 12: ‘the Programmed cell death protein 1(PD-1)/Programmed death-ligand 1 (PD-L1) pathway’ should be re-written as ‘the programmed cell death protein (PD-1)/programmmed death-ligand 1 (PD-L1) pathway’. There is no need to format these in bold.
2. Line 56: ‘Helicobacter pylori’: no need to format these in bold.
3. Line 250: ‘Ju X.et al.’ should be written as ‘Ju et al.’. Ju is a last name.
4. Line 321: TNF-a should be written as TNF-α.
5. Line 337: ‘3.83% and 0.54% respectively’ should be written as ‘3.83% and 0.54%, respectively’.
6. Reference section: Author should consult and peruse carefully recent issues of the journal, Cancers, for format and style. Also double-check the abbreviations of journal names.
Overall, the manuscript can be considered to publication after minor revision as indicated above.
Author Response
We thank the reviewer for the useful feedback. Please see the uploaded file for a detailed response.
